# Adipocyte PHLPP2 inhibition prevents obesity-induced fatty liver

KyeongJin Kim [1,2,3✉], Jin Ku Kang[1], Young Hoon Jung[2,3], Sang Bae Lee [4], Raffaela Rametta[5], Paola Dongiovanni[5], Luca Valenti[5] & Utpal B. Pajvani [1✉]

Increased adiposity confers risk for systemic insulin resistance and type 2 diabetes (T2D), but mechanisms underlying this pathogenic inter-organ crosstalk are incompletely understood. We find PHLPP2 (PH domain and leucine rich repeat protein phosphatase 2), recently identified as the Akt Ser473 phosphatase, to be increased in adipocytes from obese mice. To identify the functional consequence of increased adipocyte PHLPP2 in obese mice, we generated adipocyte-specific PHLPP2 knockout (A-PHLPP2) mice. A-PHLPP2 mice show normal adiposity and glucose metabolism when fed a normal chow diet, but reduced adiposity and improved whole-body glucose tolerance as compared to Cre- controls with high-fat diet (HFD) feeding. Notably, HFD-fed A-PHLPP2 mice show increased HSL phosphorylation, leading to increased lipolysis in vitro and in vivo. Mobilized adipocyte fatty acids are oxidized, leading to increased peroxisome proliferator-activated receptor alpha (PPARα)-dependent adiponectin secretion, which in turn increases hepatic fatty acid oxidation to ameliorate obesity-induced fatty liver. Consistently, adipose PHLPP2 expression is negatively correlated with serum adiponectin levels in obese humans. Overall, these data implicate an adipocyte PHLPP2-HSL-PPARα signaling axis to regulate systemic glucose and lipid homeostasis, and suggest that excess adipocyte PHLPP2 explains decreased adiponectin secretion and downstream metabolic consequence in obesity.

[1] Department of Medicine, Columbia University, New York, NY, USA. [2] Department of Biomedical Sciences, College of Medicine, Inha University, Incheon, South Korea. [3] Department of Biomedical Science, Program in Biomedical Science and Engineering, Inha University, Incheon, Republic of Korea. [4] Division of Life Sciences, Jeonbuk National University, Jeonju, Republic of Korea. [5] Internal Medicine and Metabolic Diseases, Fondazione IRCCS Ca' Granda Ospedale Maggiore Policlinico, DEPT, Università degli Studi di Milano, Milano, Italy. ✉email: kimkj@inha.ac.kr; up2104@columbia.edu

Obesity prevalence is increasing worldwide[1,2] and is associated with an imbalance in anabolic and catabolic processes. For example, triglyceride (TG) synthesis and storage are increased in multiple tissues in obesity; this lipogenic process is opposed by lipolysis, an insulin-regulated process in white adipose tissue (WAT), which liberates fatty acid and glycerol[3,4]. Lipolysis is further associated with catabolic remodeling by increasing fat oxidation, leading WAT to resemble a brown adipose tissue (BAT)-like phenotype[5–7].

Lipolysis is under tight hormonal control according to the nutrient state of the organism[8,9], In fasting, β3-adrenergic receptor (β3-AR)-protein kinase A (PKA) signaling promotes phosphorylation of hormone-sensitive lipase (HSL) and perilipin, leading to increased lipolysis. In the postprandial state, insulin action leads to reduced lipolysis[10], but mechanisms by which key lipolytic enzymes are regulated in periods of high energy demands are not fully understood.

Pleckstrin Homology Domain Leucine-Rich Repeat Protein Phosphatases (PHLPPs), PHLPP1 and PHLPP2, dephosphorylate Akt Ser473 and have well-described tumor-suppressive properties[11–16]. Far less is understood about the role of PHLPPs in metabolic regulation. Recently, we found decreased PHLPP2 levels in the obese liver, which leads to unrestrained insulin action, excess de novo lipogenesis, and fatty liver[17,18]. During these studies, we observed far higher expression of PHLPP2 in adipose than the liver or other tissues[17]; here, we find that adipocyte PHLPP2 is further increased in obese mice, suggesting that PHLPP2 may play an important role in adipocyte biology. Consistent with this hypothesis, adipocyte PHLPP2 ablation increases adipose lipolysis due to prolonged HSL phosphorylation. Surprisingly, excess lipolysis is not detrimental to overall glucose and lipid homeostasis—in fact, adipocyte-specific PHLPP2 KO (A-PHLPP2) mice showed mildly improved glucose homeostasis, as well as a significant reduction in hepatic triglycerides, due to increased adipocyte PPARα activity, increased adiponectin secretion and hepatocyte fatty acid oxidation. This signaling axis translates to patients as well, as adipose PHLPP2 levels negatively correlate with serum adiponectin. Overall, these data suggest that blocking excess adipocyte PHLPP2 may uncouple obesity from its metabolic comorbidities.

## Results

### Blocking adipocyte PHLPP2 attenuates HFD-induced adipocyte hypertrophy.

PHLPP1 and 2 proteins are highly expressed in white adipose tissue (WAT) and brown adipose tissue (BAT) depots[17]. Both PHLPPs are expressed preferentially in adipocytes as compared to a stromovascular fraction (SVF) in epididymal (visceral) or inguinal (subcutaneous) WAT (Supplementary Fig. 1a). Intriguingly, PHLPP2 but not PHLPP1 levels were markedly increased high-fat diet (HFD)-fed WT mice and 8-week-old leptin receptor-deficient db/db mice (Fig. 1a). Similarly, we observed that visceral adipose PHLPP2 was positively associated with BMI in obese patients undergoing bariatric surgery (Supplementary Fig. 1b), with the caveat that there were no "lean" subjects given the nature of the experimental cohort.

To determine the repercussions of increased adipose PHLPP2 in obesity, we crossed PHLPP2 "floxed" mice[17] with Cre transgenic mice driven by the tamoxifen-inducible adiponectin promoter[19] on a C57BL/6J background (henceforth A-PHLPP2 mice, Supplementary Fig. 1c). Phlpp2 was efficiently depleted in adipose from A-PHLPP2 mice, specifically in adipocytes, without a compensatory increase of Phlpp1 (Supplementary Fig. 1d and Fig. 1b). We observed no differences in body weight and adiposity when Cre- and A-PHLPP2 mice were fed a normal chow diet (NCD). When challenged with HFD, however, A-PHLPP2 mice showed a mild reduction in body weight (Fig. 1c) and adiposity

(Fig. 1d), attributable to reduced WAT and liver weight (Fig. 1e). Expression of canonical adipogenic or lipogenic transcription factors and markers were comparable between Cre- and A-PHLPP2 mice (Supplementary Fig. 2a, b), and adipogenesis was unaffected in PHLPP2 knockdown (PHLPP2 sg) 3T3-L1 cells (Supplementary Fig. 2c–e). However, we observed that HFD-fed A-PHLPP2 mice showed lower adipose triglyceride (TG) content (Fig. 1f), and a shift towards smaller adipocytes in both eWAT and iWAT (Fig. 1g), suggesting that PHLPP2 deficiency prevents HFD-induced adipocyte hypertrophy.

### Adipocyte-specific PHLPP2 ablation improves glucose homeostasis.

While NCD-fed A-PHLPP2 mice showed no obvious differences in glucose tolerance and insulin sensitivity as compared to Cre- controls, loss of PHLPP2 prevented progressively impaired glucose tolerance and insulin sensitivity in HFD-fed mice (Fig. 2a, b). Consistent with improved insulin sensitivity, A-PHLPP2 mice showed lower plasma insulin than Cre- control mice (Fig. 2c). Based on these data, and the known role of PHLPP2 to dephosphorylate Akt at Ser473, we predicted that improved systemic glucose homeostasis arose through increased Akt phosphorylation at that residue. Surprisingly, we observed comparable adipose Akt Ser473 phosphorylation to control littermates (Supplementary Fig. 2f), as well as in PHLPP2 knockdown 3T3-L1 adipocytes (Supplementary Fig. 2g). These data suggested previously unrevealed functions of adipocyte PHLPP2 beyond Akt dephosphorylation that impact systemic metabolism.

### Adipocyte-specific PHLPP2 ablation increases energy expenditure and fat lipolysis.

We next investigated the mechanism of decreased body weight and adiposity in the absence of adipocyte PHLPP2. We found that A-PHLPP2 mice had normal food intake (Supplementary Fig. 3a), but a trend towards increased energy expenditure in the basal state without a change in locomotor activity (Supplementary Fig. 3b–e). Consistently, pair-feeding did not affect a difference in body weight of HFD-fed A-PHLPP2 mice (Fig. 3a). Increased oxygen consumption was accentuated by concomitant β3-adrenergic agonist (CL316,243) administration to stimulate lipolysis (Fig. 3b, c). Consistent with these data, A-PHLPP2 mice showed a trend towards lower RER (Supplementary Fig. 3f). In addition, HFD-fed A-PHLPP2 mice showed comparable serum TG, cholesterol, and non-esterified fatty acid (NEFA) to control mice, but higher free glycerol levels in the fasted state (Supplementary Fig. 4a–d). Consistently, CL316,243 treatment increased plasma glycerol and NEFA in A-PHLPP2 mice as compared to Cre- mice (Fig. 3d). We also found significantly higher rates of glycerol and fatty acid release in ex vivo fat explants from A-PHLPP2 mice (Fig. 3e).

To test whether increased fat mobilization in A-PHLPP2 mice was cell-autonomous, we performed in vitro lipolysis assays in PHLPP2 knockdown 3T3-L1 adipocytes. As compared to control cells, PHLPP2 knockdown adipocytes showed greater forskolin- or CL316,243-induced glycerol and fatty acid secretion in medium (Fig. 3f). Conversely, glycerol and fatty acid release were significantly lower in CL316,243-treated, PHLPP2-transduced 3T3-L1 adipocytes (Fig. 3g and Supplementary Fig. 4e). These results indicate that increased adipocyte PHLPP2 in obesity inhibits lipolysis.

### PHLPP2 directly regulates HSL phosphorylation and localization.

We next examined the mechanism by which PHLPP2 ablation affects lipolysis. Key lipolytic genes—adipose triglyceride lipase (Atgl), hormone-sensitive lipase (Hsl), comparative gene identification-58 (CGI-58), or G0/G1 switch 2 (G0s2)[3,10]—were unchanged in eWAT and iWAT of A-PHLPP2 mice (Supplementary Fig. 5a, b), suggesting

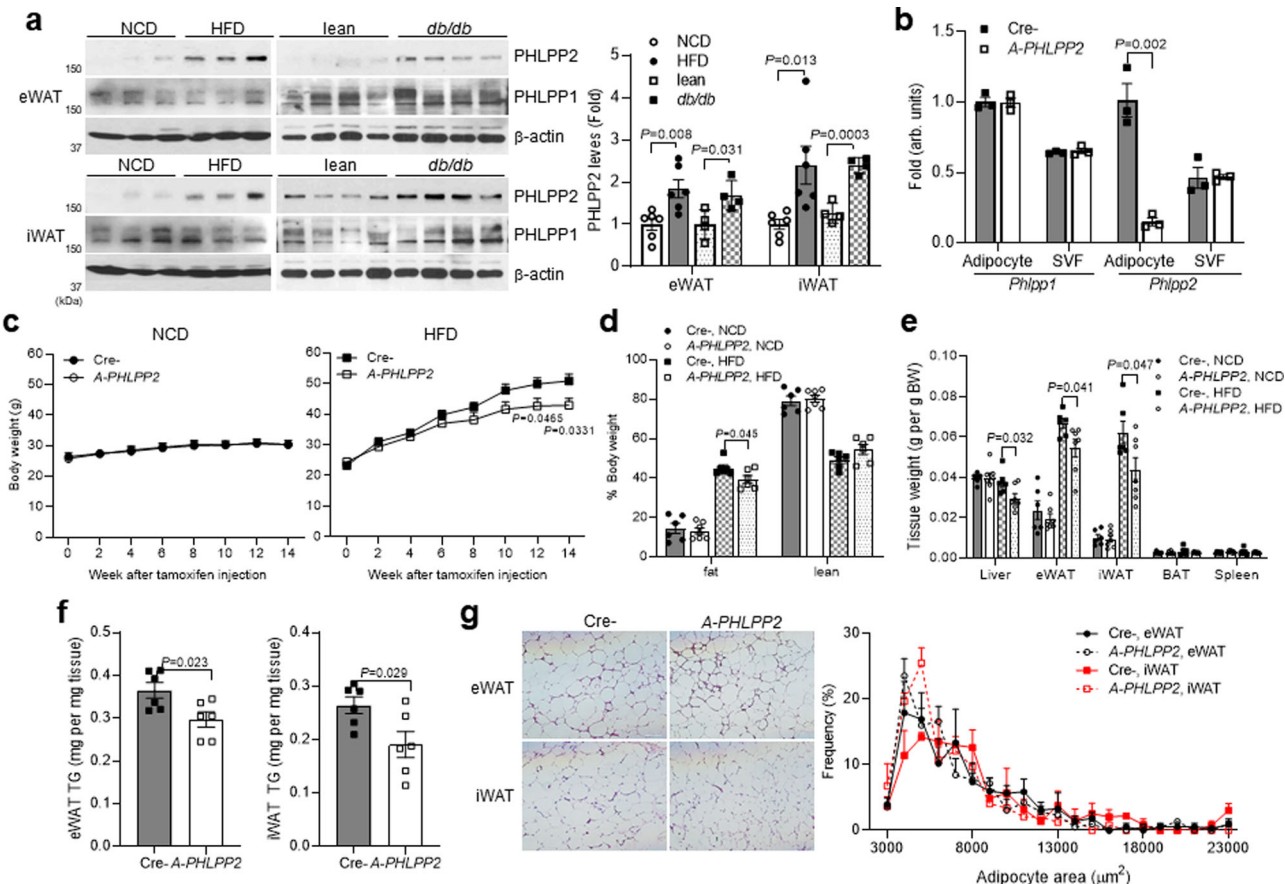

**Fig. 1 Blocking adipocyte PHLPP2 attenuates HFD-induced adipocyte hypertrophy. a** PHLPP1 and 2 protein levels in eWAT and iWAT from normal chow diet (NCD) or high-fat diet (HFD)-fed C57BL/6 J, and lean or *db/db* mice, with quantification of PHLPP2 levels (right, *n* = 4–6 independent mice per group). **b** Gene expression in adipocytes and SVF isolated from eWAT from HFD-fed Cre- and *A-PHLPP2* mice sacrificed after an 18 h fast (*n* = 3 independent mice per group). **c** Body weight curves, **d** body composition, and **e** tissue weights in NCD-fed or HFD-fed Cre-*PHLPP2* or *A-PHLPP2* mice (*n* = 6–7 independent mice per group). **f** eWAT and iWAT triglyceride (TG) content (*n* = 6 independent mice per group), and **g** H&E staining (left) and adipocyte cell size-frequency distribution (right) (*n* = 3 independent mice per group) in HFD-fed Cre- and *A-PHLPP2* mice. Scale bars, 100 μm. All data are shown as the means ± SEM. Statistical significance was determined by unpaired two-tailed Student's *t*-test.

a post-transcriptional effect. As PHLPPs are Ser/Thr phosphatases[16], we investigated the phosphorylation state of key lipolytic proteins including ATGL, HSL, or Perilipin. Of these, we observed that loss of PHLPP2 partially rescued HFD-induced defects in HSL activating phosphorylations at PKA sites Ser563 and Ser660[20,21] in both eWAT and iWAT (Fig. 4a). AMPK-induced HSL Ser565 phosphorylation, thought to be inhibitory[21], was unchanged, consistent with unaltered AMPKα T172 phosphorylation (Fig. 4a). PHLPP2 effects on HSL phosphorylation were cell-autonomous, as PHLPP2 knockdown augmented and PHLPP2 overexpression inhibited CL316,243-mediated HSL Ser563 and Ser660 phosphorylation in 3T3-L1 adipocytes (Fig. 4b, c). Interestingly, β3-adrenergic signaling, as indicated by phosphorylation of CREB and phosphor-PKA substrates (Fig. 4a, b), was unaffected by PHLPP2. These data suggest that loss of PHLPP2 does not directly alter HSL activation, but may prevent HSL dephosphorylation. To test this, we performed an in vitro phosphatase assay, using purified PHLPP2 protein. We found, as hypothesized, that PHLPP2 dephosphorylated HSL at Ser563 and Ser660 (Fig. 4d).

These data show that HSL is a direct target of PHLPP2. We next tested whether PHLPP2 regulates HSL function. Consistent with prior data[22,23], we observed translocation of HSL from cytoplasm to the surface of large lipid droplets (LDs) and micro-LDs in response to CL316,243 in control cells (Fig. 4e and Supplementary Fig. 5c). But HSL remained diffusely cytoplasmic

or at the peripheries of the LDs in PHLPP2-transduced adipocytes (Fig. 4e and Supplementary Fig. 5c). Combining these observations, we concluded that PHLPP2-mediated HSL dephosphorylation decreased HSL activity.

Next, to directly test whether altered HSL function may explain PHLPP2 effects on lipolysis, we performed both genetic and pharmacologic tests. First, we treated PHLPP2 knockdown adipocytes with a non-selective HSL inhibitor, [4-(5-methoxy-2-oxo-1,3,4-oxadiazol-3(2H)-yl)-2-methylphenyl]-carbamic acid, phenylmethyl ester (CAY10499)[24], which was able to prevent increased β3-AR-stimulated lipolysis due to knockdown of PHLPP2 (Fig. 4f). In parallel, we created 3T3-L1 adipocytes expressing HSL sgRNAs and HSL/PHLPP2 sgRNAs (Supplementary Fig. 5d) and found that HSL knockdown similarly reversed PHLPP2 effects on β3-AR-stimulated lipolysis (Fig. 4g). These findings indicate that PHLPP2 effects on lipolysis are dependent on HSL phosphorylation and activity.

**Loss of adipocyte PHLPP2 increases adipocyte oxidative machinery.** β3-AR-induced lipolysis supplies fuel for oxidative metabolism[25,26], in part by the action of the PPAR nuclear receptors[27]. Of these, PPARα and PPARδ have been shown to be required for the induction of fatty acid oxidation (FAO) and oxidative phosphorylation in brown adipocytes[28]. As serum NEFA levels were unchanged in *A-PHLPP2* mice (Supplementary

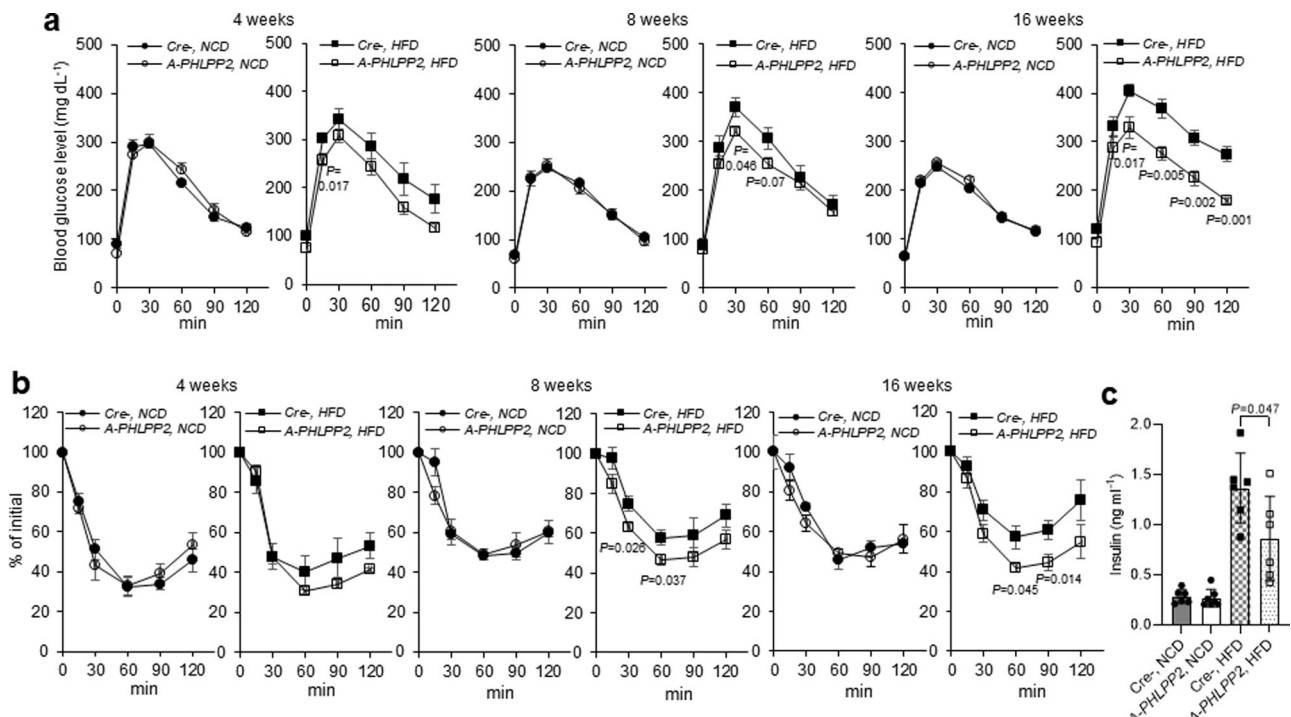

**Fig. 2 Adipocyte-specific PHLPP2 ablation improves glucose homeostasis. a** Glucose tolerance test (GTT) and (**b**) insulin tolerance test (ITT) with increasing lengths of NCD or HFD feeding in *A-PHLPP2* and control mice ($n = 6$–7 independent mice per group) after tamoxifen injection. **c** Plasma insulin levels in NCD-fed or HFD-fed *A-PHLPP2* and control mice after a 5 h fast ($n = 6$–7 independent mice per group. All data are shown as the means ± SEM. Statistical significance was determined by unpaired two-tailed Student's *t*-test.

Fig. 4c), despite induction of lipolysis by PHLPP2 ablation, we hypothesized that PHLPP2 deletion induced WAT FAO. Indeed, *A-PHLPP2* eWAT and iWAT showed increased gene expression of FAO machinery, including *PPARα*, *Acox*, and *Cpt2* (Fig. 5a). These data held true in vitro as well – PHLPP2 overexpression repressed CL316,243-stimulated *PPARα*, *Acox*, and *Cpt2* expression (Fig. 5b), whereas PHLPP2 knockdown 3T3-L1 adipocytes showed the opposite (Fig. 5c). As predicted from our molecular studies, this effect was HSL-dependent and blocked by CAY10499 treatment or HSL knockdown (Fig. 5c, d). These results suggest that PHLPP2-mediated lipolysis is linked to changes in FAO, which explains unchanged serum NEFA despite increased lipolysis in *A-PHLPP2* WAT. Indeed, the molar ratio of NEFA to glycerol released from WAT explants (Fig. 5e) or differentiated 3T3-L1 cells (Fig. 5f) was uncoupled by PHLPP2 knockdown, suggesting increased utilization of NEFA within adipocytes. To test this, we measured fatty acid oxidation in adipose from Cre- control and *A-PHLPP2* mice, by assessment of complete oxidation of $^{14}$C-oleate to $^{14}$CO$_2$. Indeed, adipose explants from *A-PHLPP2* mice showed higher FAO than Cre- controls (Fig. 5g).

**Adipocyte PHLPP2 deficiency protects from diet-induced fatty liver.** Increased adipose FAO may lead to less systemic lipid distribution. Accordingly, *A-PHLPP2* livers were smaller (Fig. 1e) with fewer lipid droplets and decreased TG content (Fig. 6a, b) but unchanged liver protein and cholesterol levels (Supplementary Fig. 6a, b). *A-PHLPP2* mice showed increased hepatic FAO gene expression (Fig. 6c), suggesting altered adipokine levels. Of these, adiponectin induces hepatic FAO through activation of AMPK and PPARα[29]. Intriguingly, we found increased adipocyte *Adipoq* expression in *A-PHLPP2* eWAT and iWAT (Fig. 6d and Supplementary Fig. 7a, b), and commensurately increased plasma adiponectin (Fig. 6e and Supplementary Fig. 7c). We compared these data to those for leptin, another adipokine, expression, and

plasma levels of which trended lower, consistent with lower fat mass in *A-PHLPP2* mice (Supplementary Fig. 7d, e). *Adipoq* expression and adiponectin protein levels were also higher in PHLPP2 knockdown 3T3-L1 adipocytes (Fig. 6f and Supplementary Fig. 7f). Conversely, adiponectin secretion was lower in 3T3-L1 adipocytes with PHLPP2-overexpression (Fig. 6g). Increased *Adipoq* expression with loss of PHLPP2 was abolished by treatment with the PPARα antagonist, GW6471 or PPARα knockdown (Supplementary Fig. 7f, g, h), suggesting that increased *Adipoq* expression depends on PPARα activity.

We next considered whether increased adiponectin was causal to higher hepatic FAO in *A-PHLPP2* mice. To test this, we treated primary hepatocytes isolated from WT mice with serum from *A-PHLPP2* or Cre- controls; with this assay, we found that FAO gene expression was increased in response to serum from *A-PHLPP2* mice (Fig. 6h), accompanied by increased phosphorylation of AMPKα and ACC1 (Fig. 6i). These effects were negated by the addition of an adiponectin neutralizing antibody (Fig. 6j and Supplementary Fig. 7i), supporting the conclusion that increased FAO gene expression in livers of *A-PHLPP2* mice is mediated by adiponectin signaling.

With a view towards translation of these mouse findings, we obtained human adipose tissue and corresponding serum in obese patients undergoing bariatric surgery. We found an inverse relationship between adipose PHLPP2 (Supplementary Fig. 7j) and plasma adiponectin (Fig. 6k). These data complement findings in HFD-mice and suggest that PHLPP2 modulation of adipocyte HSL activity regulates plasma adiponectin levels, and indirectly, hepatic steatosis.

**Discussion**
These studies indicate that PHLPP2 has distinct, tissue-specific roles that affect whole-body lipid metabolism—in the liver, hepatocyte PHLPP2 plays a protective role, terminating insulin-induced Akt signaling to prevent excess lipogenesis[18]; conversely, mice lacking

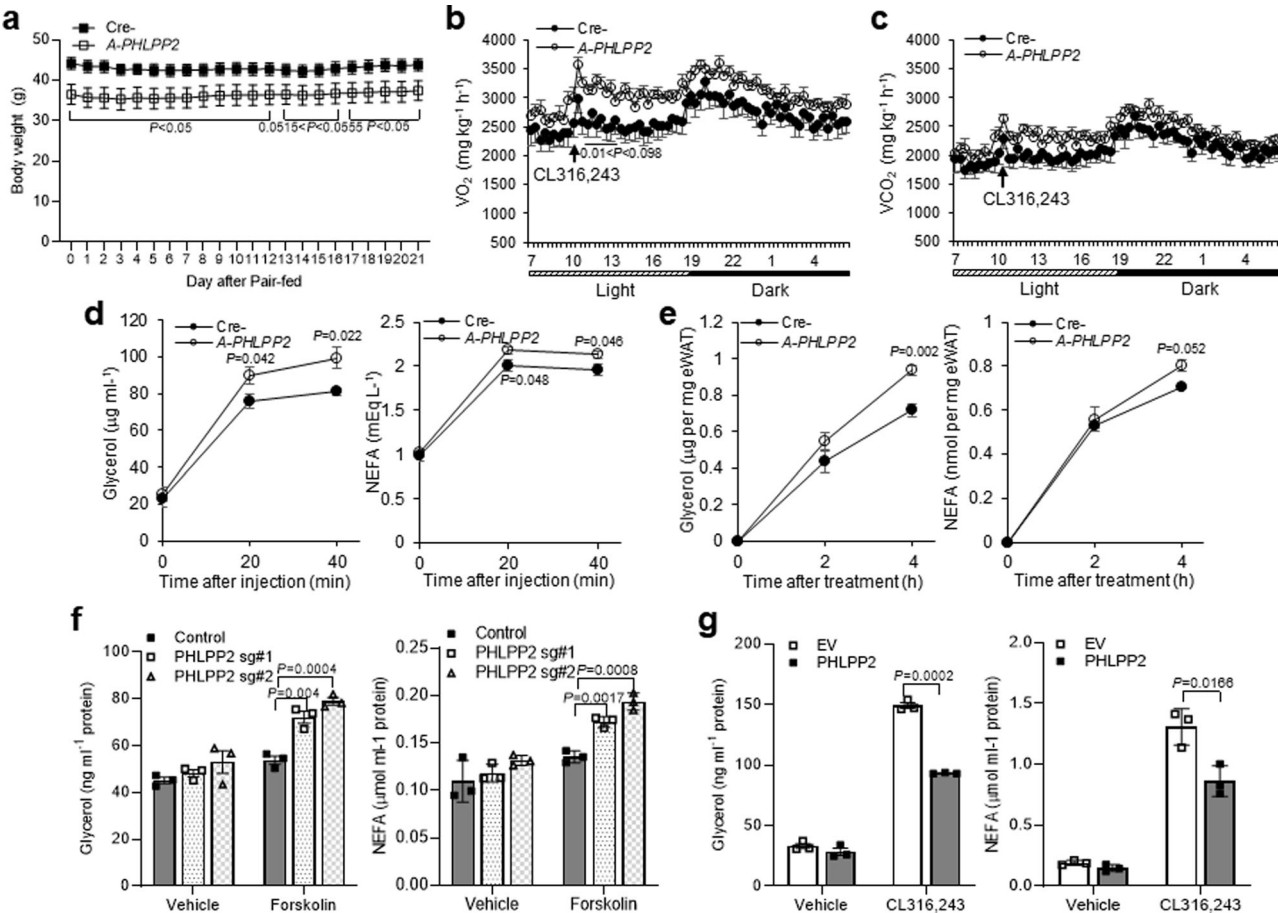

**Fig. 3 Adipocyte PHLPP2 regulates energy expenditure and fat lipolysis. a** Body weight curves in HFD-fed Cre- and A-PHLPP2 mice after initiation of pair-feeding ($n = 7$–8 independent mice per group). **b, c** $VO_2$ consumed and $VCO_2$ generated, as measured by indirect calorimetry in HFD-fed Cre- and A-PHLPP2 mice in response to the β3-adrenergic agonist CL316,243 (0.5 mg kg$^{-1}$) ($n = 6$–7 independent mice per group). **d** Serum glycerol and non-esterified fatty acid (NEFA) levels in mice following injection of CL316,243 at 0.5 mg kg$^{-1}$ ($n = 6$ independent mice per group). **e** Ex vivo lipolysis assay in isolated eWAT from Cre-PHLPP2 and A-PHLPP2 mice following treatment with 1 μM CL316,243 ($n = 3$ independent mice per group). **f** In vitro lipolysis in differentiated control or PHLPP2-repressed (using two different sgRNA sequences) 3T3-L1 adipocytes, with or without 10 μM forskolin ($n = 3$ independent experiments). **g** In vitro lipolysis in differentiated 3T3-L1 adipocytes expressing either empty vector (EV) or PHLPP2 with or without 1 μM CL316,243. ($n = 3$ independent experiments). All data are shown as the means ± SEM. Statistical significance was determined by unpaired two-tailed Student's $t$-test.

adipocyte PHLPP2 are protected from diet-induced obesity and fatty liver (Fig. 6l). We also observe opposite regulation of endogenous PHLPP2 levels across these cell types. Chronic activation of glucagon-PKA signaling in obesity prompts hepatocyte PHLPP2 protein degradation[17], but parallel increase in catecholamine-induced PKA action actually increases adipocyte PHLPP2 in diet-induced or genetic mouse models of obesity. Our data in patient-derived specimens from bariatric surgery, although limited by the absence of lean subjects, suggests similar regulation in humans. This disconnect suggests a secondary regulation of PHLPP2. One possible explanation is that the PHLPP2 3'-untranslated region harbors complementary binding sites for a subset of miRNAs belonging to the miRNA-17-92 cluster; in lymphoma cells, PHLPP2 protein was reduced by overexpression of miR-17-92[30] whereas inhibition of miR-17-92 showed the opposite[31]. Intriguingly, this cluster was shown to be repressed in human omental adipose tissues of obese, T2D patients[32,33], but this potential mechanism requires further investigation.

Loss of adipocyte PHLPP2 increased lipolysis, but serum NEFA was not increased, suggestive of balanced release from adipocytes and uptake by peripheral tissues, fatty acid oxidation, or reutilization by adipocytes through a futile re-esterification cycle. Of these, the former seems unlikely given lower hepatic TG. Indeed, lower

supernatant NEFA:glycerol ratio in adipose tissue explants from A-PHLPP2 mice supports increased intracellular utilization of fatty acids and/or rapid NEFA re-uptake. Increased oxygen consumption and heat production in A-PHLPP2 mice suggests oxidation of these lipids, which may, in turn, explain reduced adiposity and whole-body glucose homeostasis in A-PHLPP2 mice.

Increased lipolysis in A-PHLPP2 mice depends on HSL phosphorylation and activity, establishing HSL as a heretofore unidentified substrate for this phosphatase. But to clarify, it does not appear that PHLPP2 can override upstream activating signals of HSL—i.e., fasting-induced β3-AR action—but rather prolongs endogenous activity. Also of note, despite clear evidence for PHLPP2 as an Akt Ser473 phosphatase, we did not observe changes in Akt phosphorylation in A-PHLPP2 mice, or PHLPP2 gain-of-function and loss-of-function in 3T3-L1 adipocytes. However, as acute induction of lipolysis by stimulation with β3-AR agonist simultaneously promotes insulin secretion[34], Akt Ser473 phosphorylation may already be maximal in this condition. Thus, it remains possible that PHLPP2 has multiple substrates in adipocytes, rather than substrate competition with preferential dephosphorylation of HSL over Akt in the obese state.

A further intriguing aspect of these data is that adipose tissue PHLPP2 regulates hepatic TG content, by an increase in

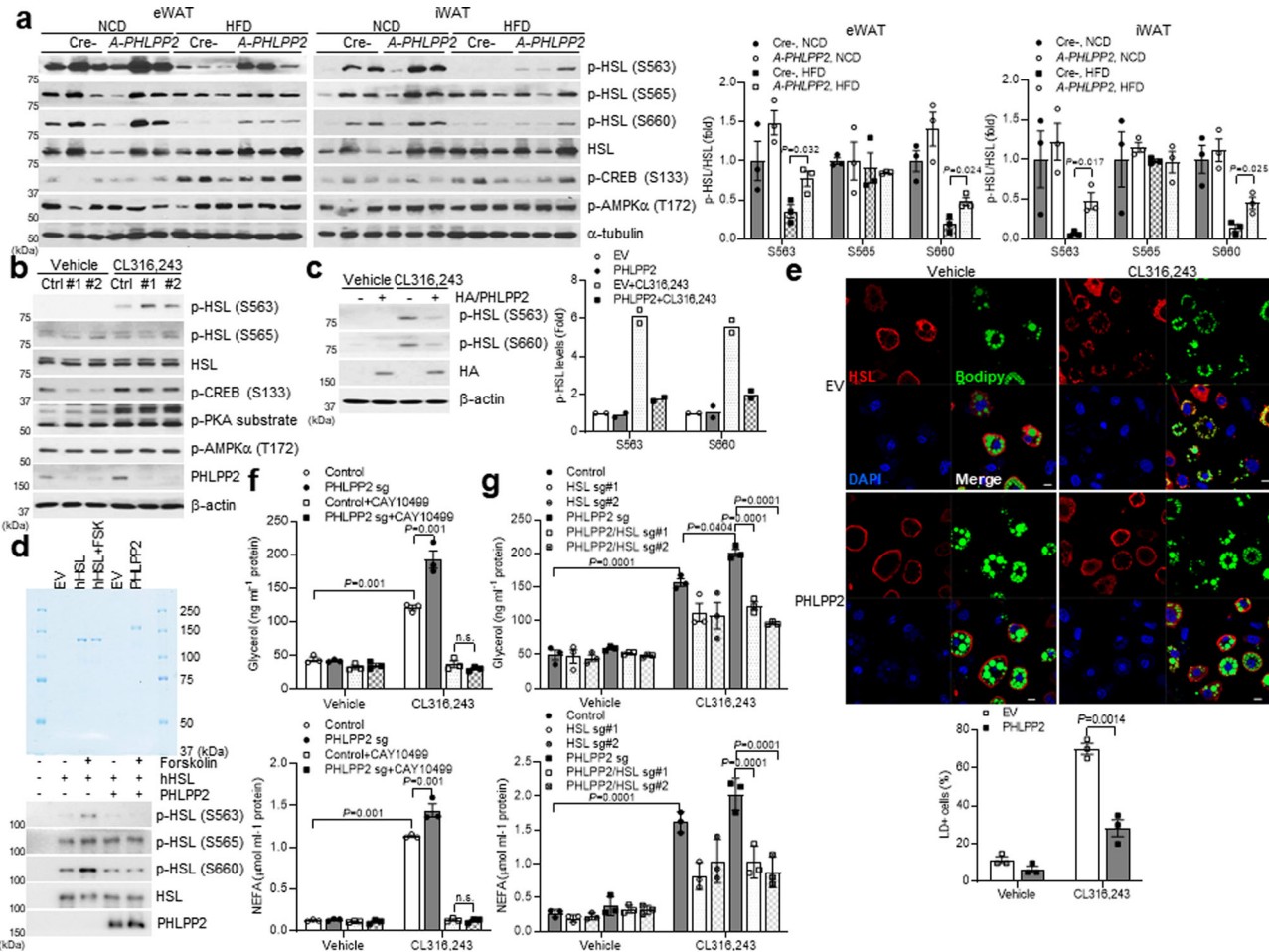

**Fig. 4 Adipocyte PHLPP2 regulates HSL phosphorylation and localization. a** Western blots in lysates from eWAT (top) or iWAT (bottom) from Cre- or *A-PHLPP2* mice fed on NCD or HFD, with quantification of various HSL phosphorylation sites (right) (n = 3 independent mice per group). **b, c** Western blots in lysates from differentiated PHLPP2-repressed 3T3-L1 adipocytes (**b**), or 3T3-L1 adipocytes expressing HA-tagged PHLPP2 (**c**) with quantification of p-HSL levels (right), with or without CL316,243 (n = 2 independent experiments). **d** Coomassie brilliant blue staining of purified human HSL (hHSL) or PHLPP2 with or without 10 μM forskolin (FSK) (top), for phosphatase assay of HSL Ser563 and Ser660 with purified PHLPP2 (bottom) (n = 2 independent experiments). **e** Immunofluorescence of HSL (red), Bodipy (green), and DAPI (blue) in 3T3-L1 adipocytes expressing control or HA-tagged PHLPP2 with or without CL316,243 (top). Quantification of HSL localization on lipid droplets was performed by directly counting Bodipy-positive cells from images of randomly chosen fields (bottom) (n = 3 independent samples). Scale bars, 10 μm. **f, g** In vitro lipolysis in differentiated control or PHLPP2-repressed adipocytes with or without CL316,243 and/or CAY10499 (**f**) or HSL deletion (**g**) (n = 3 independent mice per group). All data are shown as the means ± SEM. Statistical significance was determined by unpaired two-tailed Student's t-test for **a, e**, and two-way ANOVA with Tukey's multiple comparison test for **f, g**. n.s. not significant (P > 0.05).

*Adipoq* expression and secretion. These data may explain the well-documented reduction of plasma adiponectin in obesity[35–37], by means of a PHLPP2-HSL-PPARα axis that regulates *Adipoq* expression. We find that adipose PHLPP2 levels are negatively correlated with serum adiponectin levels in obese patients, consistent with the observation that adiponectin predicts HSL activity in patients[38]. When coupled to our findings from PHLPP2 loss-of-function mice, these data suggest that adipocyte PHLPP2 regulates systemic lipid and glucose homeostasis, and if targeted in an adipose-specific manner[39], for example, a targeted nanoparticle approach[40] encapsulating the small molecule PHLPP2 inhibitor[41,42], may present a therapeutic approach for obesity-induced metabolic abnormalities.

## Methods

**Animal**. We crossed AdipoqCreER^T2 (C57BL/6 background)[19] and PHLPP2^flox/flox (C57BL/6 background) mice[17] to generate AdipoqCreER^T2; PHLPP2^flox/flox (*A-PHLPP2*) mice. 7 to 8-week-old male *A-PHLPP2* were treated with i.p. injections of 1 mg tamoxifen dissolved in corn oil (or vehicle) for 5 consecutive days. All mice have housed 3–5 animals per cages (standard), with a 12 h light/dark cycle, in a temperature (22 °C) and humidity (40–60%)-controlled environment and were fed normal chow (Purina Mills 5053) or high-fat diet (18.4% calories/carbohydrates, 21.3% calories/protein and 60.3% calories/fat derived from lard; Research Diets Inc, D12492). All animal experiments were conducted in accordance with institutional guidelines and regulations and approved by the Columbia University Institutional Animal Care and Utilization Committee.

**Isolation of stromal vascular cells and an adipocyte-rich fraction of adipose tissues**. Epididymal or inguinal adipose tissues were isolated from mice and minced into fine (<10 mg) pieces. Minced samples were placed in a digestion solution containing type I collagenase from Worthington Biochemical Corporation (Lakewood, NJ) for 1 h at 37 °C with gentle agitation. Samples were isolated by filtration through a cell strainer and the suspension was centrifuged at 500 × g for 5 min. The pelleted cells were collected as the SVC and the top layer of lipid-containing cells were collected as an adipocyte-rich fraction.

**Lentivirus production and cell culture studies**. Lentivirus expressing PHLPP2, HSL, or PPARα single-guide RNAs (sgRNAS) were generated as previously described[17]. The complete open reading frame of PHLPP2 was amplified from pcDNA3/HA/Flag/PHLPP2[17] with PCR and cloned into the pLVX-Puro vector carrying the HA epitope tag. Viral supernatant was applied to 3T3-L1 cells with sequabrene (Sigma, St. Louis, MO), which were then selected with puromycin (ThermoFisher, Waltham, MA) before the expansion of single clones. 3T3-L1

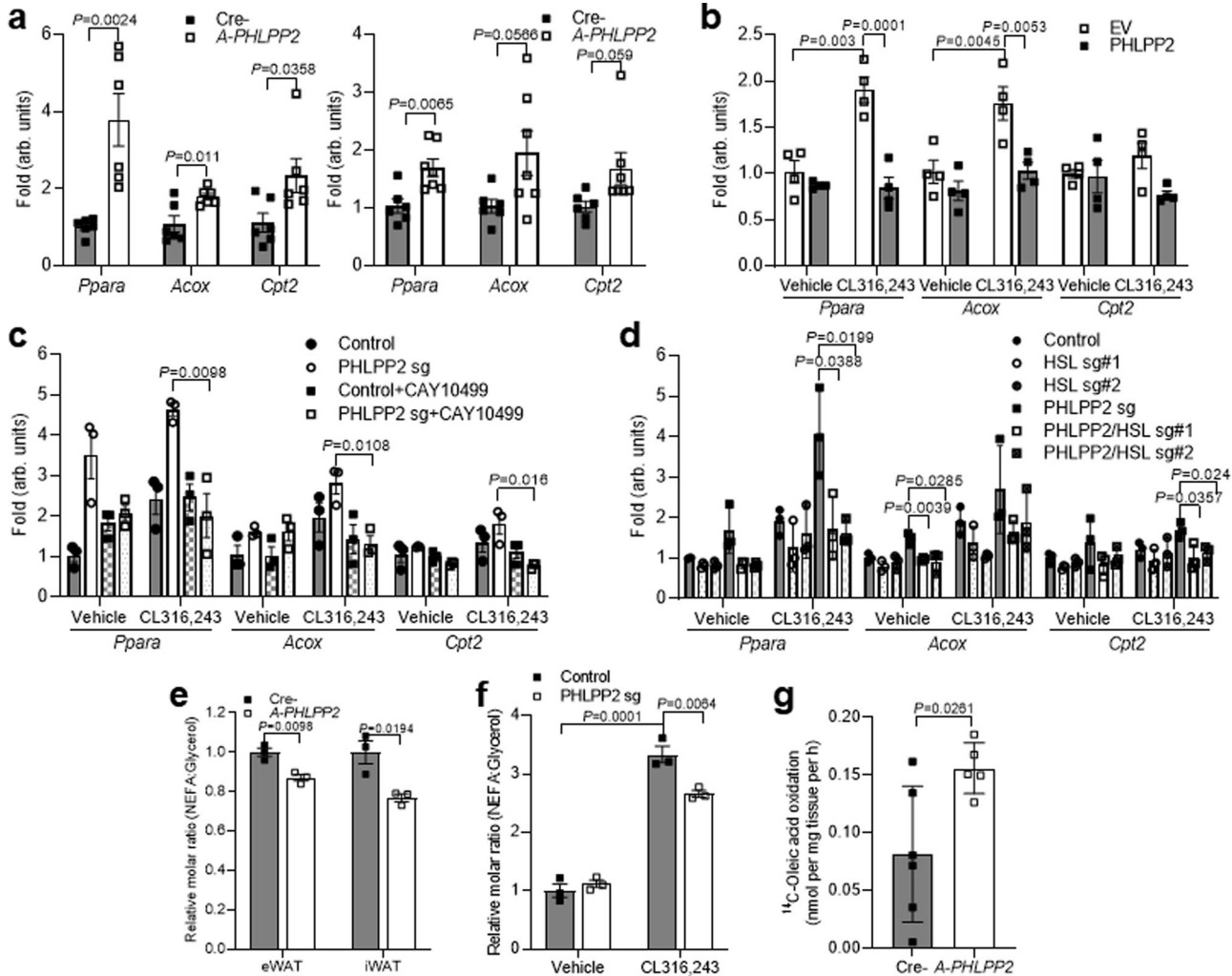

**Fig. 5 Loss of adipocyte PHLPP2 induces fatty acid oxidative machinery. a** Gene expression in eWAT (left) and iWAT (right) of HFD-fed Cre- and *A-PHLPP2* mice sacrificed after an 18 h fast ($n = 6$–7 independent mice per group). **b**–**d** Gene expression in differentiated 3T3-L1 adipocytes with PHLPP2 overexpression ($n = 4$ independent mice per group). **b** or PHLPP2 sg with or without CL316,243 and/or CAY10499 (**c**) or HSL knockdown (**d**) ($n = 3$ independent mice per group). **e, f** Relative molar ratio of NEFA to glycerol release from eWAT or iWAT explants from HFD-fed Cre-*PHLPP2* and *A-PHLPP2* mice treated with 1 μM CL316,243 for 4 h ($n = 3$ independent mice per group) (**e**) or differentiated control or PHLPP2-repressed 3T3-L1 adipocytes with or without CL316,243 ($n = 4$ per group) ($n = 3$ independent mice per group) (**f**). **g** FAO of $^{14}$C-oleate to $^{14}$CO$_2$ in eWAT from Cre- and *A-PHLPP2* mice ($n = 5$–6 independent mice per group). All data are shown as the means ± SEM. Statistical significance was determined by unpaired two-tailed Student's *t*-test for **a, c**–**e, g**, and two-way ANOVA with Tukey's multiple comparison test for **b, f**.

preadipocytes were grown in DMEM supplemented with 10% calf serum. Two days after confluence, adipocyte differentiation was initiated with the addition of 1 μg ml⁻¹ insulin, 0.5 mM IBMX, and 1 μM dexamethasone in DMEM media supplemented with 10% FBS for 2 days, followed by 2 days in medium supplemented with insulin and then cultured for 4 days in the normal growth medium. Primary mouse hepatocytes were isolated as previously described[17,18,43].

**In vitro phosphatase assay.** Full-length human HSL (hHSL) cDNA was amplified from pLenti6.3/V5-DEST/HSL (DNASU) by PCR and subcloned into EcoRI and XbaI sites on the pLVX1.1 vector carrying a FLAG epitope tag. FLAG-tagged PHLPP2 or hHSL were, respectively, expressed in HEK293T cells with or without 10 μM forskolin for 1 h and purified with EZview™ Red anti-FLAG M2 affinity gel (Sigma; F2426). Purified proteins were eluted from beads in a buffer containing 50 mM Tris-HCl, pH 7.5, 150 mM NaCl, and 500 ng/ml 3X FLAG peptide (Sigma; F4799). Dephosphorylation reactions were performed with purified FLAG-PHLPP2 and hHSL as a substrate in a reaction buffer containing 50 mM Tris-HCl, pH 7.5, 1 mM DTT, and 5 mM MnCl₂ at 30 °C for 60 min.

**Metabolic analyses.** Blood tail vein glucose was measured using a glucose meter (Bayer, Leverkusen, Germany). Glucose tolerance tests or insulin tolerance tests were performed by intraperitoneal injection of 1 or 2 g kg⁻¹ body weight glucose or 0.5 IU kg⁻¹ body weight insulin after 16-hour or 5-hour fasting, respectively. Plasma triglyceride (ThermoFisher), Cholesterol E, nonesterified fatty acid (Wako,

Osaka, Japan), free glycerol (Sigma) were measured using colorimetric assays according to the manufacturer's protocol. Plasma insulin (Mercodia, Uppsala, Sweden) and adiponectin (R&D system, MN) were assessed by ELISA. Hepatic and adipose lipids were extracted by the Folch method, and measured by colorimetric assays as above. Mice were individually housed in metabolic chambers (CLAMS, Columbus Instrument) in order to assess metabolic activity. Mice were acclimated for 48 h before data were collected. Mice had free access to food and water for recordings. Oxygen consumption, carbon dioxide production, and RER were calculated at 20 min intervals[44].

**Fatty acid oxidation.** eWAT (total of ~100 mg) was minced into fine (<10 mg) pieces and then incubated in serum-free medium with 1.5% fatty acid-free BSA containing 0.1 mM cold oleic acid and $^{14}$C-oleic acid (1 μCi/ml) for 3 h. Complete oxidation of $^{14}$C-oleate to $^{14}$CO$_2$ was measured as previously described[45].

**Antibodies and western blots.** Immunoblots were conducted on 3 to 7 samples randomly chosen within each experimental cohort with antibodies against HSL (#4107 or #18381), p-HSL (S563) (#4139), p-HSL (S565) (#4137), p-HSL (S660) (#4126), Perilipin-1 (#9349), p-AMPKα (T172) (#2535), AMPKα (#2532), p-CREB (S133) (#9198), CREB (#9197), p-(Ser/Thr) PKA substrate (#9621), HA-tag (#3724), Adiponectin (#2789) and β-actin (#4970) from Cell Signaling (Denvers, MA); PHLPP1, (A300-660A), PHLPP2 (A300-661A) from Bethyl Laboratories, Inc (Montgomery, TX); PPARα (SC-9000) from Santa Cruz Biotechnology (Dallas,

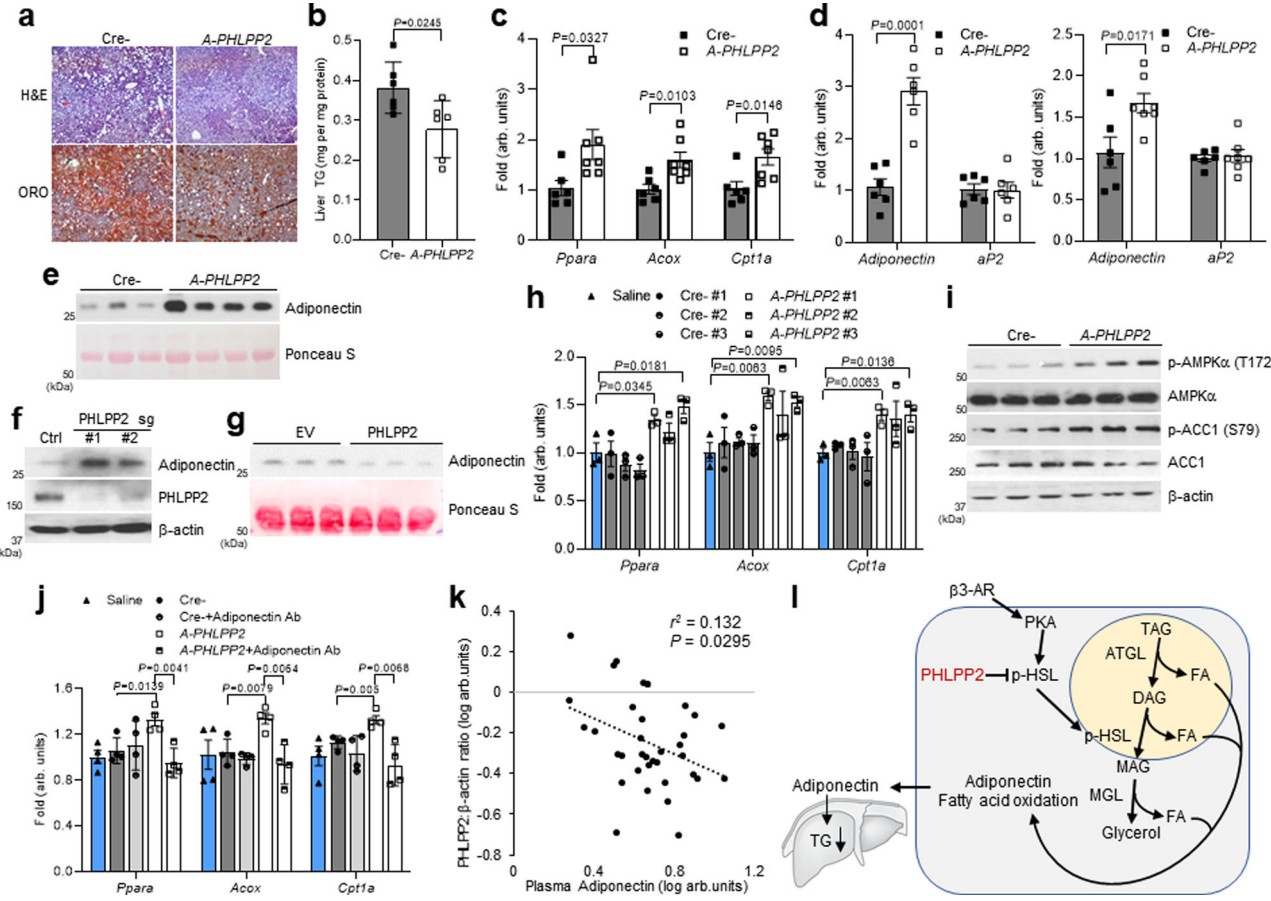

**Fig. 6 Blocking adipocyte PHLPP2 protects from diet-induced fatty liver. a–c** H&E and oil-red O (ORO) staining (Scale bars, 100 μm) (**a**), TG (**b**), and gene expression (**c**) from livers of HFD-fed Cre- and *A-PHLPP2* mice (*n* = 6–7 independent mice per group). **d, e** *Adipoq* and *aP2* gene expression in eWAT (left) or iWAT (right) (**d**), and representative serum adiponectin (**e**) from HFD-fed Cre- and *A-PHLPP2* mice (*n* = 6–7 independent mice per group). **f** Western blots in control and PHLPP2-repressed 3T3-L1 adipocytes. **g** Western blots of conditioned medium from control and PHLPP2-overexpressing 3T3-L1 adipocytes. **h** Hepatocyte gene expression in response to saline, or serum from Cre- or *A-PHLPP2* mice (*n* = 3 independent samples per group). **i** Western blots in primary hepatocytes treated with serum from Cre- or *A-PHLPP2* mice (*n* = 3 independent samples). **j** Hepatocyte gene expression in response to saline, or serum from Cre- or *A-PHLPP2* mice with or without anti-Adiponectin neutralizing antibody (*n* = 4 independent samples per group). **k** Plasma adiponectin is inversely correlated to adipose PHLPP2:β-actin ratio, as analyzed by linear regression, with the $r^2$ and *P* values indicated (*n* = 36 per independent samples). **l** Model representing the effect of adipocyte PHLPP2 on lipolysis to regulate whole-body glucose homeostasis and hepatic lipid accumulation. All data are shown as the means ± SEM. Statistical significance was determined by unpaired two-tailed Student's *t*-test.

TX). Mouse serum was neutralized by an adiponectin antibody (ab3455) from Abcam[46]. All primary antibodies were used at a dilution of 1:1000. HRP-linked anti-rabbit IgG (NA934) and anti-mouse IgG (NXA931) were purchased from Amersham and used at a dilution of 1:5000. Alexa Fluor 594 anti-Rabbit IgG (A32754) was purchased from ThermoFisher and used at a dilution of 1:1000.

**In vivo, ex vivo, and in vitro lipolysis assays**. For the in vivo lipolysis assay, blood was collected 0, 10, and 20 min after β3-adrenergic agonist CL316,243 (Tocris Bioscience) i.p. injection into mice. For ex vivo or in vitro lipolysis assays, 20 mg of epididymal, inguinal adipose tissue explants or differentiated 3T3-L1 cells were cultured in Krebs-ringer HEPES buffer (115 mM NaCl, 5.9 mM KCl, 1.2 mM MgCl₂, 1.2 mM NaH₂PO₄, 2.5 mM CaCl₂, 25 mM NaHCO₃, 12 mM HEPES, pH 7.4) with 0.5% fatty acid-free BSA (Sigma) and 5 mM glucose. The medium was collected 0, 60, and 120 min after adding 1 μg/ml CL316,243, with levels of FFA and glycerol normalized to the weight of adipose explants or protein concentrations from 3T3-L1 cell lysates.

**Quantitative RT-qPCR**. RNA was isolated by TRIzol (Invitrogen, Carlsbad, CA) or RNeasy lipid tissue mini kit (Qiagen), and cDNA synthesized with High-Capacity cDNA Reverse Transcription kit (Applied Biosystems, Foster City, CA), followed by quantitating reverse-transcriptase PCR with Power SYBR Green PCR master mix (Applied Biosystems) in a CFX96 Real-Time PCR detection system (Bio-Rad, Hercules, CA). Primer sequences are available in Supplementary Table 1.

**Quantification and statistical analysis**. To assess statistical significance, we performed an unpaired two-tailed *t*-test for comparison of 2 groups, or ANOVA followed by unpaired two-tailed *t* or Turkey tests for studies involving multiple groups. All data are shown as mean ± SEM. Sample size and statistical details can be found in the figure legends.

**Reporting summary**. Further information on research design is available in the Nature Research Reporting Summary linked to this article.

## Data availability

All data are available from the corresponding authors upon reasonable request. Source data are provided with this paper.

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

## Acknowledgements

We thank Ana Flete and Thomas Kolar, Jun Feranil, Alicja Skowronski, and Victor Lin (Columbia University Transgenic Core Facility) for advice and excellent technical support, Li Qiang, Baeki E. Kang, and Dongryeol Ryu for insightful discussion, Jinsook Son and Liheng Wang for providing adipose samples and Feng Zhang for the LentiCRISPRv2 plasmid. This work was supported by NIH DK103818 (UBP), an INHA UNIVERSITY Research Grant (K.K.), and National Research Foundation of Korea grants funded by the Korean government (MSIT) (No. 2020R1C1C1004015 for K.K. and No. 2020R1C1C1014281 for S.B.L).

## Author contributions

K.K. designed, performed, and interpreted experiments, and wrote the manuscript. J.K.K., Y.H.J., and S.B.L. performed experiments. U.P. designed and interpreted experiments and wrote the manuscript. R.R., P.D., and L.V. provided human adipose tissue and corresponding serum. All authors critically read the manuscript.

## Competing interests

The authors declare no competing interests.
