## [Peer Review File · Nature Communications]

Reviewers' comments:

Reviewer #1 (Remarks to the Author):

The function of PHLPP2 is quite novel in current study. The authors overextend their results to suggest the mechanism underlying PHLPP2 regulation of p-HSL which are not fully supported by the data. My specific concerns are below.

1. As a phosphatase, PHLPP2 exhibits its function through targeting multiple substrates. To discover the deep mechanisms underlying the effects of PHLPP2 on lipolysis, the direct substrate of PHLPP2 should be identified in this study.

2. The authors claimed that "we did not observe changes in Akt phosphorylation in A-PHLPP2 mice, or PHLPP2 gain- and loss-of-function in 3T3-L1 adipocytes." But the results of p-Akt (S473) in 3T3-L1 adipocytes were not shown in the manuscript. More important, to confirm PHLPP2 did not change in Akt phosphorylation, the authors should supplement the results of total AKT expression in both mouse and 3T3-L1 adipocytes.

3. The authors have to show more light-dark cycles for the data generated by metabolic cages. Also, it is likely that the mice were not well controlled by the circadian clock, as it can be proved by the fact that both food uptake and energy expenditure were not dramatically altered between light and dark. Moreover, an altered O_2/CO_2 ratio is required to confirm the increased FAO in ko mice, however, by checking the fig3a and b, an increased O_2/CO_2 was observed in ko mice, indicating that glucose oxidation rather than FAO is a favored carbon source for energy expenditure. Are activities differed between ko and wt mice? Heat data in fig3c cannot be used to make a conclusion on thermogenesis. Browning of WAT or activation of BAT must be test before the conclusion of thermogenesis can be made.

4. In Figure 1D, the four groups were difficult to follow.

Reviewer #2 (Remarks to the Author):

Kim, Pajvani and colleagues investigated the role of the phosphatase PHLPP2 in adipocytes. In vivo, they studied adipocyte-specific PHLPP2 knockout mice. In vitro, they manipulated PHLPP2 levels in 3T3-L1 adipocytes. They also provide data on adipose PHLPP2 expression in mice and humans. In conclusion, the author postulates that they characterized a novel pathway involving PHLPP2, the neutral lipase HSL and the nuclear receptor PPARalpha and suggests that the pathway regulates adiponectin secretion.

Main points

1. One of the main conclusions is that that excess adipocyte PHLPP2 observed in obesity contributes to decreased adiponectin secretion and downstream metabolic consequences.

1.1 The claim that PHLPP2 expression is increased in obesity and related metabolic disorders needs to be better substantiated. Fig. 1c shows that the level of expression in the stromavascular fraction is not negligible compared to adipocyte level (Fig. 1c, Supp Fig. 1a). The number of mice reported in Fig. 1a and b is low. There is no statistical analysis. No data are reported in humans. This point may be addressed through Western blot analyses of isolated adipocytes in several mouse models of obesity, insulin resistance and (pre)diabetes. BXD mouse strains used in their 2017 Gastroenterology article on hepatic PHLPP2 are a fruitful resource in that respect. Large existing biobanks of adipose tissue, sometimes with corresponding isolated adipocyte fractions, may be probed for human adipose PHLPP2 (mRNA if not protein) expression.

1.2 The paper shows that decreased expression of PHLPP2 in adipocytes contributes to enhanced adiponectin secretion. There is no demonstration, besides correlative data in a rather small human cohort, showing that increasing adipose PHLPP2 results in lower adiponectin levels. In vitro, it can be measured whether there is, as postulated, decreased secretion of adiponectin in culture media of adipocytes overexpressing PHLPP2 (Fig. 4c and Supp Fig. 4f). In that respect, 3T3-L1 adipocytes may not be the most suitable model as adiponectin expression levels are low.

1.3 Measurement of leptin adipose gene expression and plasma levels in adipose PHLPP2 knockout and control mice fed high fat diet may provide further proof of changes in fat mass (Fig. 1d-f) and highlight the specificity of the major upregulation of adiponectin (compared to another adipokine, leptin) plasma levels (Fig. 6e). On the latter aspect, data on 3 and 4 mice are shown in Fig. 6e and not on the 7-8 mice per group mentioned in Figure legend which have to be reported.

1.4 The demonstration that adiponectin is under a pathway involving adipose PHLPP2 / lipolysis-derived fatty acids / PPARalpha activation in white adipocyte is weak (Supp Fig. 7c). In vitro, selectivity of PPARalpha antagonist in white adipocytes expressing high amounts of PPARgamma is questionable. In vivo, the role of PPARalpha in white adipocytes seems rather subtle (Lasar et al. Cell Rep. 2018 22:760-773). Further evidence may be provided by direct manipulation of PPARalpha levels in vitro and/or in vivo.

2. Impact of adipose PHLPP2 on hormone-sensitive lipase (HSL) and lipolysis requires clarification and additional experimental data.

2.1 Immunofluorescence of HSL in 3T3-L1 adipocytes expressing PHLPP2 suggests that translocation of HSL from cytosol to the surface of lipid droplets following adrenergic stimulation is impaired in adipocytes overexpressing PHLPP2 (Fig. 4d). This is an important piece of data.

Several independent experiments with various microscopic fields need to be shown as Supp data. Lipolysis (i.e., glycerol and NEFA levels in culture media) has to be measured in parallel.

2.2 CAY10499 is used as an inhibitor of HSL (Fig. 4e, 5c). However, this compound is a potent inhibitor of adipose triglyceride lipase and monoacylglycerol lipase, the two other lipases involved in fat cell lipolysis (Chembiochem. 2008 9:2704-10; J Lipid Res. 2016 57:131-41). This explains why complete inhibition of NEFA release is observed in Fig. 4e. Partial inhibition should be observed when selectively inhibiting HSL. Therefore, it cannot be used to demonstrate that the specific involvement of HSL in PHLPP2-mediated effect on lipolysis and fatty acid oxidation. Specific inhibitors of HSL are available.

2.3 Adipose PHLPP2 shows selectivity in dephosphorylating substrates. Activating phosphorylation sites of HSL (Ser 563 and Ser660) but not HSL Ser565 and CREBP Ser133 are dephosphorylated (Fig. 4 a-c). Akt Ser473 which is a target in the liver is not in adipocytes (SuppFig2d). Mechanistic clue is lacking. The authors discuss the potential involvement of miRNA or other PHLPP2 substrate than Akt in adipocytes but do not provide data.

Other points

1. On top of what is detailed above, the number of biological replicates is too low and/or no statistical analysis is reported for several sets of data (notably Fig. 3a-c, Fig. 4a, Fig. 4c).

2. The following sentence in the abstract is confusing : "To study consequence of increased adipocyte PHLPP2, we generated adipocyte-specific PHLPP2 knockout (A-PHLPP2) mice." Please, rephrase.

Reviewer #3 (Remarks to the Author):

This manuscript provides interesting data on effects of PHLPP2 KO versus overexpression in adipocytes, and suggests a role in regulating basal levels of lipolysis. This is associated with changes in p-HSL levels, suggesting that PHLPP2 acts directly on p-HSL to dephosphorylate it and impede its stimulation of lipolysis (a main conclusion of the paper). Phenotypes of the KO mice are of interest and clearly established, especially the lower weight on HFD. Overall, this is a study of interest, but some major concerns are noted:

1. The PHLPP2 KO mice are leaner on HFD than WT mice, but the effects on EE and on food intake are not conclusive, as both seem to show trends. Since both EE and food intake are difficult to measure in mice, and small changes often missed, it is possible that much of the in vivo phenotype of the mice are simply due to lower food intake and lower weight (which would secondarily increase insulin sensitivity and glucose tolerance). It is therefore difficult to tease out cause and effect when comparing mice of different weights, especially since loss of calories through excretion was not apparently measured. Pair feeding these mice would be necessary to definitively discover the cause of the weight loss on HFD. If the mice were still lower weight during pair feeding, it would indicate that EE or change in excretion of calories were the reason. In this study it seems that a clear conclusion on the reason for the loss of weight on HFD is needed to make firm conclusions.

2. Another major deficiency of the manuscript is the lack of data on a key point of the paper: what is the direct substrate for dephosphorylation by PHLPP2? Is HSL a direct substrate, and does that explain the phenotype? In vitro cell free studies should be done to show this. Also, in the last sentence of the first paragraph on page 7, the authors state that loss of PHLPP2 does not affect activation of HSL, but rather prolongs the effect of PKA based on the data with PKA substrate antibodies. However, this conclusion is not rigorous, as it is probably the overall balance between phosphorylation and dephosphorylation at all times that determines HSL activity.
3. Fig 4d shows only 1 field each for the different conditions studied, and therefore is not valid. These data need to be quantified by signal determination in many fields in a doubly blinded manner, and a bar graph of the results presented in the paper with statistical analysis.
4. Since in liver PHLPP2 seems to modulate de novo lipogenesis, is this the case in adipocytes?
5. Related to verifying the A-PHLPP2 mouse model, the gene expression in Fig 1c was decreased by about 80%, while the reduction in Supplementary 1c was only about 40%. Please clarify which Figure illustrates eWAT/iWAT, and please give explanations for the lower knockdown results.
6. If increased adipocyte fatty acid oxidation is the trigger for subsequent events in the PHLPP2 mice, it would be important to show that fatty acid oxidation is actually increased by measuring metabolic flux rather than just enzyme mRNA. Also, determining RER would give an indication if higher levels of fatty acid oxidation is the case in the KO mice.

We appreciate the thoughtful comments and suggestions of editors and reviewers, and believe that our revised manuscript has been greatly improved by the modifications made in response to their comments. Below, we provide a point-by-point discussion of these comments and questions.

Reviewers' comments:

Reviewer #1 (Remarks to the Author):

The function of PHLPP2 is quite novel in current study. The authors overextend their results to suggest the mechanism underlying PHLPP2 regulation of p-HSL which are not fully supported by the data. My specific concerns are below.

We appreciate the reviewer's comments and hope that we have addressed the key mechanistic point raised.

1. As a phosphatase, PHLPP2 exhibits its function though targeting multiple substrates. To discover the deep mechanisms underlying the effects of PHLPP2 on lipolysis, the direct substrate of PHLPP2 should be identified in this study.

We agree – although we saw increased HSL phosphorylation at Ser563 and 660 in PHLPP2 knockout adipocytes, and decreased phosphorylation at these sites with

overexpression of PHLPP2, these data do not prove that HSL is a direct PHLPP2 substrate, a point also raised by Reviewer #3. To address this critical point, we performed an *in vitro* phosphatase assay using purified proteins (detailed explanation in the revised Methods section) which establishes that PHLPP2 directly dephosphorylates HSL at Ser563 and 660 (**new Fig. 4d**).

2. The authors claimed that “we did not observe changes in Akt phosphorylation in A-PHLPP2 mice, or PHLPP2 gain- and loss-of-function in 3T3-L1 adipocytes.” But the results of p-Akt (S473) in 3T3-L1 adipocytes were not shown in the manuscript. More important, to confirm PHLPP2 did not change in Akt phosphorylation, the authors should supplement the results of total AKT expression in both mouse and 3T3-L1 adipocytes.

We apologize for this omission. In **new Supplementary Fig. 2e and g**, we now show both p-Akt and total Akt levels in Cre- and A-PHLPP2 adipose, as well as control and PHLPP2-deficient 3T3-L1 cells. These data suggest that loss of PHLPP2 does not affect Akt S473 phosphorylation in adipose.

3. The authors have to show more light-dark cycles for the data generated by metabolic cages. Also, it is likely that the mice were not well controlled by the circadian clock, as it can be proved by the fact that both food uptake and energy expenditure were not dramatically altered between light and dark. Moreover, an altered O₂/CO₂ ratio is required to confirm the increased FAO in KO mice, however, by checking the fig3a and b, an increased O₂/CO₂ was observed in KO mice, indicating that glucose oxidation rather than FAO is a favored carbon source for energy expenditure. Are activities differed between KO and WT mice? Heat data in fig3c cannot be used to make a conclusion on thermogenesis. Browning of WAT or activation of BAT must be tested before the conclusion of thermogenesis can be made.

In the revised version of this manuscript, we provide more light-dark cycle data, showing clearly altered food intake, energy expenditure and activities between light and dark (**new Supplementary Fig. 3a-e**). To address the FAO point raised, we directly measured oxidation of ¹⁴C-oleate to ¹⁴CO₂ in WAT from control and KO mice (**new Fig. 5g**). Increased FAO in A-PHLPP2 mice is consistent with the trend toward decreased RER in KO mice (**new Supplementary Fig. 3f**). But as to the reviewer's final point, we absolutely agree. Although loss of adipocyte PHLPP2 may affect thermogenesis, WAT browning or BAT activation, we cannot conclude this with the data in hand.

4. In Figure 1D, the four groups were difficult to follow.

We separated the graphs into chow (left) and HFD (right) groups now (**new Fig. 1c**).

Reviewer #2 (Remarks to the Author):

Kim, Pajvani and colleagues investigated the role of the phosphatase PHLPP2 in adipocytes. In vivo, they studied adipocyte-specific PHLPP2 knockout mice. In vitro, they manipulated PHLPP2 levels in 3T3-L1 adipocytes. They also provide data on adipose PHLPP2 expression in mice and humans. In conclusion, the author postulates that they characterized a novel pathway involving PHLPP2, the neutral lipase HSL and the nuclear receptor PPARalpha and suggests that the pathway regulates adiponectin secretion.

Main points

1. One of the main conclusions is that that excess adipocyte PHLPP2 observed in obesity contributes to decreased adiponectin secretion and downstream metabolic consequences.

1.1 The claim that PHLPP2 expression is increased in obesity and related metabolic disorders needs to be better substantiated. Fig. 1c shows that the level of expression in the stromavascular fraction is not negligible compared to adipocyte level (Fig. 1c, Supp Fig. 1a). The number of mice reported in Fig. 1a and b is low. There is no statistical analysis. No data are reported in humans. This point may be addressed through Western blot analyses of isolated adipocytes in several mouse models of obesity, insulin resistance and (pre)diabetes. BXD mouse strains used in their 2017 Gastroenterology article on hepatic PHLPP2 are a fruitful resource in that respect. Large existing biobanks of adipose tissue, sometimes with corresponding isolated adipocyte fractions, may be probed for human adipose PHLPP2 (mRNA if not protein) expression.

We appreciate the reviewer's suggestions, and hope that the experiments presented below provide more clarity. In Fig. 1a and b, we presented Western blots from 4-6 representative samples. In the revised version of manuscript, we provide the statistical analysis for these blots (**Fig. 1a, right panel**).

To test whether this relationship exists in humans as well, we obtained adipose tissue from patients undergoing bariatric surgery. Using this precious resource, we analyzed the relationship of adipose PHLPP2 levels with BMI, which showed a positive correlation (**new supplementary Fig. 1b**). These data have limitations, as there are no "lean" controls given the nature of the experimental cohort, which we explicitly state in Discussion. But these data suggest an association of PHLPP2 levels with obesity in humans.

1.2 The paper shows that decreased expression of PHLPP2 in adipocytes contributes to enhanced adiponectin secretion. There is no demonstration, besides correlative data in a rather small human cohort, showing that increasing adipose PHLPP2 results in lower adiponectin levels. In vitro, it can be measured whether there is, as postulated, decreased secretion of adiponectin in culture media of adipocytes overexpressing PHLPP2 (Fig. 4c and Supp Fig. 4f). In that respect, 3T3-L1 adipocytes may not be the most suitable model as adiponectin expression levels are low.

We appreciate the reviewer's suggestion, and have now added data from more human samples (total 36 human adipose tissues and corresponding serum) in **new Fig. 6k** and **new Supplementary Fig. 7j**. To address the second point, we found reduced secretion of adiponectin in culture media of adipocytes overexpressing PHLPP2 in **new Fig. 6g**.

1.3 Measurement of leptin adipose gene expression and plasma levels in adipose PHLPP2 knockout and control mice fed high fat diet may provide further proof of changes in fat mass (Fig. 1d-f) and highlight the specificity of the major upregulation of adiponectin (compared to another adipokine, leptin) plasma levels (Fig. 6e). On the latter aspect, data on 3 and 4 mice are shown in Fig. 6e and not on the 7-8 mice per group mentioned in Figure legend which have to be reported.

This is an excellent point. We measured by leptin gene expression, as well as circulating levels by ELISA in control and *A-PHLPP2* mice – although differences were not significant, both were reduced ~20% in knockout animals (**new Supplementary Fig. 7d and e**), consistent with a similar reduction in fat mass seen in **Fig. 1e**.

To the reviewer's second point, in **Fig. 6e**, we showed representative western blots for adiponectin secretion. In the revised manuscript, we provide the remaining western blot results for serum adiponectin in Cre- and *A-PHLPP2* mice (**new Supplementary Fig. 7c**).

1.4 The demonstration that adiponectin is under a pathway involving adipose PHLPP2 / lipolysis-derived fatty acids / PPARalpha activation in white adipocyte is weak (Supp Fig. 7c). In vitro, selectivity of PPARalpha antagonist in white adipocytes expressing high amounts of PPARgamma is questionable. In vivo, the role of PPARalpha in white adipocytes seems rather subtle (Lasar et al. Cell Rep. 2018 22:760-773). Further evidence may be provided by direct manipulation of PPARalpha levels in vitro and/or in vivo.

The role of PPARalpha in white adipocytes is still under debate. Multiple reports have described the role of PPARalpha function in regulation of adiposity, adiponectin, and lipolysis (Guerre-Millo M et al., 2000. JBC 275: 16638-16642; Hiuge A et al., ATVB. 2007. 27: 635-641; Guzman M et al., 2004. JBC. 279: 27849-27854). The study cited by the reviewer suggests that PPARgamma, but not PPARalpha, is required for mature brown adipocyte thermogenic functionality induced by β -adrenergic signaling (using UCP1-Cre). This report cannot distinguish the PPAR isoform necessary to regulate lipolytic capacity in white adipocytes.

But we appreciated the reviewer's experimental suggestion and generated PHLPP2 KO, PPARalpha KO, and PHLPP2/PPARalpha DKO 3T3-L1 cells (**new Supplementary Fig. 7g**). *Adiponectin* mRNA was significantly reduced in PPARalpha KO or DKO cells (**new Supplementary Fig. 7h**). These data suggest PPARalpha mediates PHLPP2-regulated *Adiponectin* expression.

2. Impact of adipose PHLPP2 on hormone-sensitive lipase (HSL) and lipolysis requires clarification and additional experimental data.

2.1 Immunofluorescence of HSL in 3T3-L1 adipocytes expressing PHLPP2 suggests that translocation of HSL from cytosol to the surface of lipid droplets following adrenergic stimulation is impaired in adipocytes overexpressing PHLPP2 (Fig. 4d). This is an important piece of data. Several independent experiments with various microscopic fields need to be shown as Supp data. Lipolysis (i.e., glycerol and NEFA levels in culture media) has to be measured in parallel.

We appreciate the Reviewer's points. To clarify, the figures of immunofluorescence of HSL (**new Fig. 4e**) are representative images to show the localization of HSL. For the revised manuscript, we repeated this experiment and had an independent, blinded scientist quantitate HSL localization to lipid droplets in **new Supplementary Fig. 5c (additional representative images) and Fig. 4e (bottom)**. In addition, we measured lipolysis (glycerol and NEFA levels) in culture media in **new Fig. 3g**.

2.2 CAY10499 is used as an inhibitor of HSL (Fig. 4e, 5c). However, this compound is a potent inhibitor of adipose triglyceride lipase and monoacylglycerol lipase, the two other lipases involved in fat cell lipolysis (Chembiochem. 2008 9:2704-10; J Lipid Res. 2016 57:131-41). This explains why complete inhibition of NEFA release is observed in Fig. 4e. Partial inhibition should be observed when selectively inhibiting HSL. Therefore, it cannot be used to demonstrate that the specific involvement of HSL in PHLPP2-mediated effect on lipolysis and fatty acid oxidation. Specific inhibitors of HSL are available.

The reviewer is correct, as CAY10499 has also been shown to inhibit other lipases including ATGL, MGL and DAGLa. A selective HSL inhibitor, i.e. BAY 59-9435, was not commercially available. But as even a more specific test, we generated HSL KO, PHLPP2 KO and HSL/PHLPP2 DKO 3T3-L1 cells (**new Supplementary Fig. 5d**). Absence of HSL did not alter adipocyte differentiation, consistent with a previous report (Okazaki et al., Diabetes. 2002. 51: 3368-3375). But we found significant reduction of glycerol and NEFA release upon β 3-AR stimulation and reduced FAO gene expression in HSL KO cells; further, HSL KO reversed the increase in glycerol/NEFA release and FAO gene expression conferred by PHLPP2 KO (**new Fig. 4g and Fig. 5d**). Collectively, these results support the specific involvement of HSL in PHLPP2-mediated regulation on lipolysis and fatty acid oxidation.

2.3 Adipose PHLPP2 shows selectivity in dephosphorylating substrates. Activating phosphorylation sites of HSL (Ser 563 and Ser660) but not HSL Ser565 and CREBP Ser133 are dephosphorylated (Fig. 4 a-c). Akt Ser473 which is a target in the liver is not in adipocytes (SuppFig2d). Mechanistic clue is lacking. The authors discuss the potential involvement of miRNA or other PHLPP2 substrate than Akt in adipocytes but do not provide data.

In **new Supplementary Fig. 2e and g**, we now show p-Akt in Cre- and *A-PHLPP2* adipose, as well as control and PHLPP2-deficient 3T3-L1 cells. These data suggest that loss of PHLPP2 does not affect Akt S473 phosphorylation in adipose. But the reviewer's point is well-taken – we don't yet know why PHLPP2 (like other phosphatases) exhibit differential substrate specificity across tissues. This is an area requiring future research, but outside the scope of this report.

Other points

1. On top of what is detailed above, the number of biological replicates is too low and/or no statistical analysis is reported for several sets of data (notably Fig. 3a-c, Fig. 4a, Fig. 4c).

In the revised manuscript, we performed statistical analysis for **new Fig. 3a, b** and provide the quantification results for **Fig. 4a and c (Fig. 4a and c, bottom)**.

2. The following sentence in the abstract is confusing : “To study consequence of increased adipocyte PHLPP2, we generated adipocyte-specific PHLPP2 knockout (A-PHLPP2) mice.” Please, rephrase.

Agreed – we have rephrased this as “to identify functional consequence of increased adipocyte PHLPP2 in obese mice” in the revised manuscript.

Reviewer #3 (Remarks to the Author):

This manuscript provides interesting data on effects of PHLPP2 KO versus overexpression in adipocytes, and suggests a role in regulating basal levels of lipolysis. This is associated with changes in p-HSL levels, suggesting that PHLPP2 acts directly on p-HSL to dephosphorylate it and impede its stimulation of lipolysis (a main conclusion of the paper). Phenotypes of the KO mice are of interest and clearly established, especially the lower weight on HFD. Overall, this is a study of interest, but some major concerns are noted:

We appreciate the reviewer's comments and summary, and we hope that we have addressed the remaining points below.

1. The PHLPP2 KO mice are leaner on HFD than WT mice, but the effects on EE and on food intake are not conclusive, as both seem to show trends. Since both EE and food intake are difficult to measure in mice, and small changes often missed, it is possible that much of the in vivo phenotype of the mice are simply due to lower food intake and lower weight (which would secondarily increase insulin sensitivity and glucose tolerance). It is therefore difficult to tease out cause and effect when comparing mice of different weights, especially since loss of calories through excretion was not apparently measured. Pair feeding these mice would be necessary to definitively discover the cause of the weight loss on HFD. If the mice were still lower weight during pair feeding, it would indicate that EE or change in excretion of calories were the reason. In this study it seems that a clear conclusion on the reason for the loss of weight on HFD is needed to make firm conclusions.

We appreciate the Reviewer's suggestion – although we have not seen differences in food intake, in the revised manuscript, we performed pair-feeding as suggested in HFD-fed Cre- and *A-PHLPP2* mice. We found that PHLPP2 KO mice maintain a stable body weight deficit as compared to control mice, over 3 weeks of pair-feeding (**new Figure 3a**). This result supports the hypothesis that differences in body weight in mice lacking adipocyte PHLPP2 is likely due to increased energy expenditure.

2. Another major deficiency of the manuscript is the lack of data on a key point of the paper: what is the direct substrate for dephosphorylation by PHLPP2? Is HSL a direct substrate, and does that explain the phenotype? In vitro cell free studies should be done to show this. Also, in the last sentence of the first paragraph on page 7, the authors state that loss of PHLPP2 does not affect activation of HSL, but rather prolongs the effect of PKA based on the data with PKA substrate antibodies. However, this conclusion is not rigorous, as it is probably the overall balance between phosphorylation and dephosphorylation at all times that determines HSL activity.

We agree – although we saw increased HSL phosphorylation at Ser563 and 660 in PHLPP2 knockout adipocytes, and decreased phosphorylation at these sites with overexpression of PHLPP2, these data do not prove that HSL is a direct PHLPP2

substrate, a point also raised by Reviewer #1. To address this critical point, we performed an *in vitro* phosphatase assay using purified proteins (detailed explanation in the revised Methods section) which establishes that PHLPP2 directly dephosphorylates HSL at Ser563 and 660 (**new Fig. 4d**).

In re: the point on prolongation of HSL signaling – the Reviewer is correct. We have edited the sentence accordingly.

3. Fig 4d shows only 1 field each for the different conditions studied, and therefore is not valid. These data need to be quantified by signal determination in many fields in a doubly blinded manner, and a bar graph of the results presented in the paper with statistical analysis.

We appreciate the Reviewer's points. To clarify, the figures of immunofluorescence of HSL (**new Fig. 4e**) are representative images to show the localization of HSL. For the revised manuscript, we repeated this experiment and had an independent, blinded scientist quantitate HSL localization to lipid droplets in **new Supplementary Fig. 5c (additional representative images) and Fig. 4e (bottom)**.

4. Since in liver PHLPP2 seems to modulate de novo lipogenesis, is this the case in adipocytes?

In the revised manuscript, we provide the result for lipogenic gene expression in adipose tissues from control and *A-PHLPP2* mice (**new Supplementary Fig. 2b**), showing no differences between those groups.

5. Related to verifying the A-PHLPP2 mouse model, the gene expression in Fig 1c was decreased by about 80%, while the reduction in Supplementary 1c was only about 40%. Please clarify which Figure illustrates eWAT/iWAT, and please give explanations for the lower knockdown results.

We apologize for any confusion due to imprecise figure legends, which we have now revised. *Phlpp2* gene expression in isolated adipocytes from eWAT (**new Fig. 1b**) is down ~80, but **Supplementary Fig. 1d** shows a ~40% reduction in *Phlpp2* gene expression in intact eWAT due to preserved SVF expression.

6. If increased adipocyte fatty acid oxidation is the trigger for subsequent events in the PHLPP2 mice, it would be important to show that fatty acid oxidation is actually increased by measuring metabolic flux rather than just enzyme mRNA. Also, determining RER would give an indication if higher levels of fatty acid oxidation is the case in the KO mice.

We observed a trend toward decreased RER in KO mice (**new Supplementary Fig. 3f**). But to directly address whether FAO was different in response to loss of PHLPP2, we directly measured oxidation of ¹⁴C-oleate to ¹⁴CO₂ in WAT explants, and observed increased FAO in KO mice (**new Fig. 5g**).

REVIEWERS' COMMENTS

Reviewer #1 (Remarks to the Author):

I would like to thank to authors for their efforts and kind replies to each comment. Addition of new data certainly increased the quality of the presented work.

Reviewer #2 (Remarks to the Author):

The Authors addressed most of the comments. There are however a few points that need clarifications.

Some of the methods, materials or samples used in Supplementary Figures are not described.

Page 5 "Similarly, we observed that adipose PHLPP2 levels are positively associated with BMI in obese patients undergoing bariatric surgery (Supplementary Fig. 1b)". Which fat depot? Subcutaneous or visceral? As ratios are represented, log transformation is not required unless normality tests are shown before and after transformation. Limitations of the cohort must be stated in the main text.

Page 6 "A-PHLPP2 mice showed a trend towards lower RER, suggested that adipocyte PHLPP2 may regulate lipid utilization (Supplementary Fig. 3f)." However, there is no statistical data supporting this claim. Whereas ex vivo data suggest increased fatty acid oxidation in white adipose tissue of KO mice (Fig. 5g), no evidence of increased fatty acid oxidation is provided in vivo. This must be clearly stated.

Knock down rather than knock out shall be used to describe cell models using lentivirus expressing (and not encoding!) single-guide RNAs

Please, modify Results and Methods section to indicate that CAY10499 is a non selective lipase inhibitor.

We appreciate the thoughtful comments and suggestions of editors and reviewers, and believe that our new manuscript has been greatly improved by the modifications made in response to their comments. Below, we provide a point-by-point discussion of these comments and questions.

REVIEWERS' COMMENTS

Reviewer #1 (Remarks to the Author):

I would like to thank to authors for their efforts and kind replies to each comment. Addition of new data certainly increased the quality of the presented work.

We thank the reviews for kind comments.

Reviewer #2 (Remarks to the Author):

The Authors addressed most of the comments. There are however a few points that need clarifications.

Some of the methods, materials or samples used in Supplementary Figures are not described.

We appreciate the reviewer's comments and hope that we have addressed the remaining points below.

Page 5 "Similarly, we observed that adipose PHLPP2 levels are positively associated with BMI in obese patients undergoing bariatric surgery (Supplementary Fig. 1b)". Which fat depot? Subcutaneous or visceral? As ratios are represented, log transformation is not required unless normality tests are shown before and after transformation. Limitations of the cohort must be stated in the main text.

We analyzed visceral fat biopsied at time of bariatric surgery. In the revised manuscript, we clarified it and stated the limitation of this cohort in the main text. And we appreciate the reviewer's suggestion, as log transformation may not be required. We replaced the results with fold unit in the revised manuscript.

Page 6 "A-PHLPP2 mice showed a trend towards lower RER, suggested that adipocyte PHLPP2 may regulate lipid utilization (Supplementary Fig. 3f)." However, there is no statistical data supporting this claim. Whereas ex vivo data suggest increased fatty acid oxidation in white adipose tissue of KO mice (Fig. 5g), no evidence of increased fatty acid oxidation is provided in vivo. This must be clearly stated.

We removed the later part of this sentence in the revised manuscript.

Knock down rather than knock out shall be used to describe cell models using

lentivirus expressing (and not encoding!) single-guide RNAs

As cells expressing sgRNA were not selected as single clones, we agree and have adjusted the language accordingly.

Please, modify Results and Methods section to indicate that CAY10499 is a non selective lipase inhibitor.

We modified accordingly to indicate that CAY10499 is a non-selective HSL inhibitor.